# Integration of Network Slicing and Machine Learning into Edge Networks for Low-Latency Services in 5G and beyond Systems

**Afra Domeke** *[ID], **Bruno Cimoli** [ID] and **Idelfonso Tafur Monroy** [ID]

Department of Electrical Engineering, Eindhoven University of Technology,
5600 MB Eindhoven, The Netherlands; b.cimoli@tue.nl (B.C.); i.tafur.monroy@tue.nl (I.T.M.)
* Correspondence: a.domeke@tue.nl

**Abstract:** Fifth-generation (5G) and beyond networks are envisioned to serve multiple emerging applications having diverse and strict quality of service (QoS) requirements. To meet ultra-reliable and low latency communication, real-time data processing and massive device connectivity demands of the new services, network slicing and edge computing, are envisioned as key enabling technologies. Network slicing will prioritize virtualized and dedicated logical networks over common physical infrastructure and encourage flexible and scalable networks. On the other hand, edge computing offers storage and computational resources at the edge of networks, hence providing real-time, high-bandwidth, low-latency access to radio network resources. As the integration of two technologies delivers network capabilities more efficiently and effectively, this paper provides a comprehensive study on edge-enabled network slicing frameworks and potential solutions with example use cases. In addition, this article further elaborated on the application of machine learning in edge-sliced networks and discussed some recent works as well as example deployment scenarios. Furthermore, to reveal the benefits of these systems further, a novel framework based on reinforcement learning for controller synchronization in distributed edge sliced networks is proposed.

**Keywords:** network slicing; edge computing; machine learning; SDN controller synchronization





## 1. Introduction

Compared with existing 4G, there are significant improvements in 5G networks in terms of coverage, management, and accounting capabilities [1]. Such improvements boost the rapid pace of innovation in cellular communication technologies and generate new use cases in the era of the Internet of Things (IoT), autonomous driving, and augmented and virtual reality services. Empowered by these emerging applications and growing number of end-users, the number of interconnected devices as well as the volume of generated data have also been growing tremendously from year to year. According to the most recent Ericsson mobility report, currently there are around 8.1 billion mobile subscriptions and it is expected that total mobile data traffic will reach around 288 EB per month in 2027, meaning leading mobile networks will carry almost 300 times more traffic than in 2011 [2]. As a result, one of the main goals of 5G systems has become solving challenges that are not effectively addressed by 4G, such as demands for higher data rates, higher capacities, lower latency, real time processing, connectivity of massive numbers of devices, higher reliability, lower cost, improved QoS, and quality of experience (QoE).

As the conventional networks based on "one-size-fits-all" design are unable to address these requirements efficiently and effectively, recent efforts have sought paradigm shifts in the network architecture [3]. In this respect, the concepts of network function virtualization, software defined networking, and edge computing has appeared. These technologies are recognized as the key enablers of 5G networks as they are able to provide a scalable, flexible, and programmable network platform to manage multiple services with heterogeneous requirements within strict performance limits. All of these efforts will also enable new vertical business segments and services for consumers and enterprise customers.

## 1.1. Edge Computing

The main idea of edge computing is to extend the capabilities of cloud servers to edge of networks by performing computationally-intensive tasks and storage at the closest point of interaction to fulfil low latency requirements [4]. As the long-distance transmission of data from end devices to the cloud servers incurs a great propagation delay, bringing the computational resources to the edge of the networks that is close to Radio Access Network (RAN) and User Equipment (UE) reduces latency and provides applications with real-time performance [5]. There are also several other advantages of processing data at the edge of the networks such as mitigating the threat of data leakage by eliminating the risk of single point of failure, minimizing the risk of traffic congestion, and providing scalability to the networks which makes edge computing one of the major enabler of 5G systems [6].

In this respect, European Telecommunications Standards Institute (ETSI) formerly defined and standardized edge computing concept as Multi-Access Edge Computing (MEC) to allow the efficient and seamless integration of applications from vendors, service providers, and third-parties [7]. In their white paper, they defined the necessary specifications for MEC, provided reference architectures and recommendations, and also discussed some application scenarios such as massive sensor data analysis, active device location tracking and big mobile data. In addition to providing an execution environment for applications at the edge, MEC provides services with UE and RAN statistics, such as the radio channel quality of users and their location in the network, allowing us to build context-aware applications [8]. In particular, several 5G use cases are expected to rely on edge computing:

1. Healthcare: Edge computing in healthcare focuses on capturing, analysing, and synthesizing of necessary information by effectively prioritizing critical traffic, accelerating computer-intensive operations such as compression and decompression of medical surgery images, eliminating performance overheads, and helping to improve security [9].
2. Video analysis: A major benefit of edge computing is to allow processing of video data within the end devices such as cameras, mobile phones, or vehicles that have processing power. This enhances the transition efficiency, reduces the network bandwidth load significantly, and allows end users to make faster decisions in critical situations [10].
3. Smart home and city: Edge computing can be useful to manage and orchestrate devices for smart homes and cities by reducing response times of these devices by performing computations and data caching locally [11].

Although edge computing reduces latency, offers low operating costs, and increases consumer satisfaction, it also brings new challenges in the field of privacy and energy efficiency [12]. Enormous real-time data collection from mobile devices, and the challenge of storing and processing them in edge servers could potentially lead to a violation of the security and confidentiality of the data. In addition to that, continuous collection and transmission of data between mobile devices and edge servers consumes a huge amount of energy which introduces an energy efficiency challenge, as mobile devices are often powered by batteries. Thus, such issues should not be overlooked while designing edge sliced network solutions.

## 1.2. Network Slicing

Network slicing has been proposed to address the diversified service requirements [13]. The basic idea of network slicing is to create multiple virtual networks (i.e., slices) on top of a common physical network infrastructure to provide a flexible, centralised and programmable control and abstraction to the networks as well as eliminating the tight coupling between network functions and specific hardware units. Specifically, a network slice is a self-contained network with its own virtual resources, topology, traffic flow, and provisioning rules which gives the slice tenant the ability to operate its own dedicated physical network. These logical networks are created and managed by observing the



demands of end-users and administrators and then provided to different services to fulfil users' varying communication requirements. Therefore, network slicing is considered as one of the key enablers of 5G systems as it enables dynamic, agile, and scalable networks to respond rapidly to changing business requirements.

Software-Defined Networking (SDN) and Network Function Virtualization (NFV) are two enablers of the network slicing approach. In particular, SDN separates the control plane from the forwarding plane to offer administrators the ability to configure the network [14]. On the other hand, NFV aims to decouple network functions from physical network resources such as routers, firewalls, etc., and delivers equivalent network functionality without any specialized hardware. In this respect, there are several applications where network slicing could be beneficial:

1. Autonomous driving: Network slicing supports the performance of autonomous driving by providing mobility management, seamless continuity, and ultra-high reliable and low latency communication between the vehicles and the network, even at high speeds. In particular, creating dedicated slices for this use case can ensure that shared infrastructure absolutely does not cause any negative impact on the service operation [15].
2. Energy consumption: Having a centralized view of network, network slicing can play an important role in promptly and precisely responding to power outages by transmitting and monitoring the critical data and controlling the necessary signals/switches [16].
3. Augmented Reality (AR)/Virtual Reality (VR): AR/VR applications demand extremely low-latency as well as high resolution and bandwidth requirements to construct a virtual environment where people can have real-time interaction. In this respect, a purpose-built network slice may fulfil these heterogeneous performance requirements and ensure minimal processing overheads [15].
4. Industry 4.0: Network slicing is considered a key enabler of industry 4.0 use cases and simultaneously requires latency, reliability, device synchronicity, data rates, seamless mobility, and energy efficiency [16].

It should be noted that network slicing may also give rise to privacy and security issues [12,17]. Unlike previous generations, where only mobile operators had access and control over the system components, 5G and beyond networks allow many partners to slice network components for different use cases and vertical-specific blocks. Thus, while designing solutions with network slicing, it is important to ensure the privacy of each user and security of each slice, as well as the overall network to guarantee safe and accurate operations.

*1.3. Machine Learning*

In addition to these, machine learning (ML) is also expected to be necessary for 5G networks and the emerging use cases. Broadly speaking, an ML algorithm can analyse huge volumes of data, detect anomalies, predict future scenarios, and quickly adapt to fluctuating environments. Such functionalities allow ML to improve and automate network management within the network environment. In fact, there are a number of use cases where ML can be useful such as power-saving, fault management, maintenance, operation, power control, network configuration, QoS prediction, and throughput and performance of coverage.

There are three main categories into which ML algorithms can be divided:

1. Supervised Learning: supervised learning is useful for classification or prediction of the tasks based on a labeled dataset. In general, supervised learning has been beneficial for applications that can create large amounts of data, as the number of instances directly influences the algorithm robustness [18]. Some example application areas of supervised algorithms can be predicting network demand, coverage area, or

energy availability to dynamically allocate the resources and maximize the network performance.

2.  Unsupervised Learning: in unsupervised learning, the data used to train the algorithm is not labelled. Thus, given the data, it tries to discover subgroups with similar characteristics without any guidance. This technique is especially useful to detect patterns and relationships that may not be clearly visible in the dataset [18]. As an example, the unsupervised learning technique can be useful when detecting abnormality or fault in wireless networks, capturing correlations within the data traffic, or clustering fog nodes in heterogeneous edge networks.

3.  Reinforcement learning: reinforcement learning (RL) algorithms are especially useful in stochastic environments under uncertainty. In an uncertain environment, the system dynamics can be modelled using a Markov decision process (MDP). Then, the RL algorithm is used to find the optimal policy by trying possible actions and learning from the feedback in a given state [18]. Some application areas of RL are user scheduling, resource allocation, channel allocation, and handover decisions.

Considering the necessity of automation of network functions for design, deployment, control, and management of the networks, Ref. [19] elaborated a comprehensive overview on possible ML techniques for the network functions. Details are presented in Table 1.

**Table 1.** Network functions and relevant ML techniques [19].

| Function | ML Technique | Objective |
|---|---|---|
| Network planing, management and monitoring | <ul><li>K-means clustering;</li><li>Deep neural network;</li><li>Reinforcement Learning;</li><li>SVM.</li></ul> | <ul><li>Clustering users and service requirements;</li><li>Routing and forwarding decisions;</li><li>Resource optimization;</li><li>Parameter configuration;</li><li>Forecasting resource usage.</li></ul> |
| Fault detection and security | <ul><li>Principal component analysis;</li><li>Logistic regression;</li><li>Deep neural network.</li></ul> | <ul><li>Classification of users and applications;</li><li>Anomaly detection;</li><li>Predicting unusual behaviour.</li></ul> |

In addition to these, ML algorithms have also demonstrated significant improvements in enhancing the communication reliability for various applications such as radio resource allocation, physical security, signal decoding, and channel estimation [20,21]. Such applications require high computational power as well as accurate and efficient estimation, where ML plays a critical role. In this respect, recent studies also showed that Bidirectional LSTM, KNN, and Random Forest algorithms in particular can outperform traditional methods in terms of estimation performance and computational complexity.

### 1.4. Paper Motivation

Although such technological innovations have shown promising results, network operators still search for a better integration of 5G to unveil its full potential and consolidate 5G driven applications across multiple vertical industries. This is due to the fact that new application areas such as real-time wireless video streaming services demand multiple requirements from the use of 5G networks at the same time. As an example, it requires networking resources to broadcast video, computational resources to process it, and storage resources to locally cache it [10]. Similarly, some applications, such as ultra-high definition video and augmented reality need high-speed, high-capacity communications, while others such as the mission-critical IoT and autonomous vehicles require ultra low latency, ultra-reliable services. In this respect, a new term called edge slicing has arisen. By combining two promising approaches, SDN and edge computing, network operators aim to efficiently and simultaneously meet the diverse and multiple requirements of several vertical applications, offering flexible data transmissions and computing capabilities with minimal latency.

In this paper, our aim is to provide an extensive and comprehensive literature review of the recent research on joint edge computing and network slicing applications and integration of ML into them. Although there are previous works reviewing network slicing and edge computing mechanisms in 5G and beyond networks, our paper focuses on their combination and provides recent works on edge enabled network slicing mechanisms together with currently used ML methods. In this respect, to the best of our knowledge, this is the first review paper on edge enabled network slicing mechanisms and ML applications.

The remainder of this paper is outlined as follows. Section 2 presents a literature survey on the applications of network slicing to edge of the networks. Section 3 presents a literature survey about network slicing and edge computing with the contribution of ML techniques. Then, Section 4 describes our solution and model based on network slicing, edge computing, and RL techniques. Lastly, Section 5 concludes the paper.

## 2. Edge Enabled Network Slicing

In the current context, where there is a wide variety of services and devices that wireless networks have to deal with, both network slicing and edge computing techniques help us to satisfy QoS and QoE requirements for heterogeneous use cases. However, to achieve a complete cross-layer solution and to manage, program, and slice a heterogeneous wireless network, both paradigms seem to be necessary [22]. This is because neither network slicing nor edge computing is able to satisfy the strict timing and performance requirements of the 5G services alone [23]. Indeed, as shown in Figure 1, MEC plays different roles for different network slices [24,25]. For ultra-reliable, low-latency communication (URLLC) services, both storage and computational resources to undertake the function of data processing, analysis, and storage can be provided to ensure low latency and high reliability. In case of enhanced Mobile Broadband (eMBB) services, as high bandwidth is crucial, caching and traffic offloading at the edge can help to increase overall capacity of the mobile core. Similarly, for massive machine type communication (mMTC) applications, to provide high performance and scalability, computational and storage resources can be provided by the edge to deal with huge amounts data generated by IoT devices.

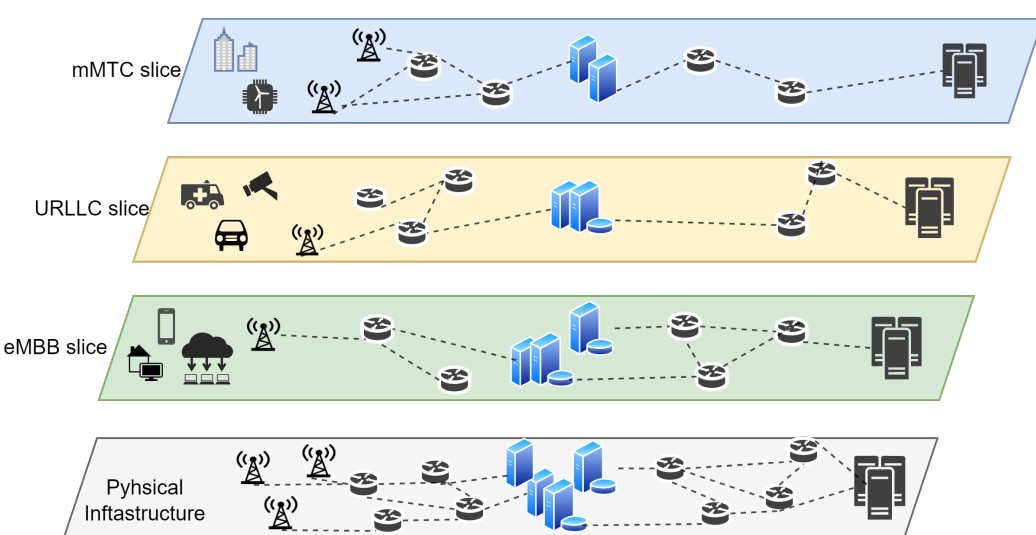

**Figure 1.** The role of edge computing and slicing for different type of services.

However, deploying and managing a sliced wireless network at the edge to achieve an effective slicing mechanism is not an easy task. This is because both computation and storage resources are very limited at the edge of networks, which makes them hard to efficiently manage. Each task requested by applications need to be satisfied and to maximize the benefit of edge slicing, one should accurately compute and analyse the requirements of slices and prioritize if necessary. Another critical challenge aroused from the fact that the wireless channel and backhaul conditions are time-varying and cannot be

discovered in advance. Therefore, before deciding how to slice an edge resource, which is a long-term decision, several things need to be considered simultaneously to guarantee the QoS of multiple differentiated services.

There are also set of challenges that sets MEC slicing apart from traditional resource allocation problems. Here, the key issue is that MEC resources are usually coupled, meaning that slicing one resource usually leads to a performance degradation in another type of resource [23]. Although algorithms that do not take isolation of slices into account can have lower time complexities compared to isolated ones, isolation among the partitions have better guarantees for QoS as well as Service Level Agreements (SLAs). Based on this observation, we classified all the studies we want to present into two categories as papers with non-isolated slices and with isolated slices.

### 2.1. Edge Enabled Network Slicing without Slice Isolation

Having these issues in mind, some frameworks for virtualizing wireless networks have been proposed in the last years. As an example, ETSI is one of the research institutes that publishes a group report on the support of network slicing in MEC systems where they present some use cases, requirements, and recommendations [26]. In their report, they first discussed different definitions and specifications of network slicing concepts of different institutes such as NGMN, ONF, 3GPP, and ETSI ISG NFV. Then, based on these concepts, they discussed some use cases on the support for MEC-enabled network slicing and provided some recommendations and key issues. This is an important work in the sense that it gathers all the existing deployments of network slicing and tries to come up with a common MEC-enabled slicing technique that is consistent with all the platforms by analysing what they currently have, what is necessary, and what should be done in the future to fully realize such a scheme.

Ref. [27] introduced a 5G enabled platform for vertical automotive industry. On the integration of NFV to the edge of the networks towards fast and reliable 5G systems, and considering critical slicing requirements of such an infrastructure, they proposed a novel slicing framework called Katana which basically is an end-to-end (E2E) slice manager. The key advantage of this architectural approach is that it offers simplicity in building and maintaining applications, flexibility, and scalability, while the containerized approach makes the applications independent of the underlying system. The results show that, the proposed framework is promising as it provides noteworthy improvement on the access time, reducing the latency by up to 9 times compared to not using it. In addition to that, they measured the scalability performance of ETSI-Open Source MANO platform which is an E2E network service orchestrator [28]. For the performance analysis, they considered slice deployment and termination times along with slice deployment time scalability which is a measure of scalability of slice deployment operations. As a result of this experiment, obtained graphs are linear which shows that OSM orchestrator has an excellent scalability. In particular, their study is promising as it shows that deployment of a NFV at the edge proves to be a pivotal step towards an acceleration in vertical industries.

Similarly, Ref. [29] focused on the benefits of edge slicing for vehicular technologies. In their work, they proposed an edge computing algorithm based on network slicing and load balancing techniques to satisfy the demands of powerful computation and large storage resources by the vehicles. As part of their solution, they used both NFV and SDN slicing techniques. That is, they utilized NFV to virtualize the data plane of the sliced edge nodes and SDN to decouple the control plane from the data plane. When compared to state-of-the-art algorithms based on how successfully they are able to manage offloading the demanded tasks from the vehicles and their processing power utilization, which is the ratio between consumed processing power and maximum processing power averaged for all nodes in the physical network, their algorithm improved the resource utilization by 48%.

Ref. [30] also focused on the management of vehicular networks by utilizing MEC and the two key techniques of network slicing. Considering the increased volume of data

traffic generated by map-assisted drivings, they proposed an edge slicing architecture to satisfy the increased number of computing and storage tasks and to avoid delays resulting from the transmission of data between MEC and the cloud servers. Here, by enabling NFV in MEC, they are able to program different functions on the server and support diversified applications. In addition, by using SDN concepts, they are able to provide a global control plane to the MEC servers and manage resources and the data traffic more efficiently. The proposed MEC-based architecture is basically a two-tier server structure in which a cloud-computing server resides in the first tier and some MEC servers reside in the second tier. Considering the availability of resources and demanded QoSs, necessary computing tasks can be processed directly on vehicles, be uploaded to MEC servers or to the cloud-computing server. For example, delay-sensitive applications, such as platooning management, dynamic HD map management, and other safety-related applications, are prioritized to be processed on MEC servers to decrease the latency. As a result of this work, they are able to achieve about 50% higher network throughput compared to other state-of-the-art schemes.

Ref. [31] also investigated network slicing for MEC systems. They realized that existing works are only focused on effective edge slicing from the perspective of mobile devices, therefore they aimed to maximize operator's revenue in a system where multiple requests had been made and no prior-knowledge of traffic distributions existed. Mainly, they developed a stochastic optimization algorithm that decides whether to accept a slice request that may need low-latency computation offloading based on sub-carrier assignments and CPU cycle frequency of the system and the amount of power allocation required to fulfil this request. The simulation results showed that their algorithm can achieve a balance between the revenue and the average delay, and can significantly increase the operator's revenue compared to existing schemes. Here, they compared their results with three main schemes which are a scheme that does not have slice request admission, a scheme that has a fixed channel allocation, and a scheme that only optimizes slice request admissions.

Similarly, Ref. [32] proposed a novel method of realizing E2E network slicing by jointly addressing the requirements of physical network and locations of users. In particular, their algorithm utilizes Dijkstra's shortest path algorithm to choose the servers with fewer bandwidth usages for the placement of VNFs and to calculate a feasible path with a minimum bandwidth consumption. Their results showed that they are able to improve the load balancing by reducing the load of edge data centres.

Ref. [33] proposed a network slicing model for 5G mobile networks, including MEC, C-RAN (Cloud radio access network), and cloud data centre to provision QoS. Mainly, they utilized queueing theory to derive some QoS performance metrics such as the CPU utilization, throughput, drop rate, average number of message requests, average response time, and average waiting time. In addition, they highlight how to use the proposed network slicing model for the dynamic scalability of C-RAN/MEC cores. The researchers also gave significant illustrations on the impact of MEC nodes on the system delay by stating that the average response time without any MEC node is approximately nine times higher than with 10 MEC nodes. However, the average time response as a function of the MEC nodes saturates after a certain threshold. The systematic outcomes and experimental processes demonstrated that MEC nodes are important to provide low latency and the proposed slicing model has a robust capability to allocate the count of MEC and C-RAN cores required to attain the quality of service targets of 5G slices.

Based on these studies, it can be seen that slicing the edge resources offer several performance and security advantages as it provides significantly lower latency as well as better scalability, coverage flexibility, and cost-efficiency. Leveraging the edge deployments, network operators can increase their operational efficiency with central slice management and can scale their solution to serve multiple customers.

### 2.2. Edge Enabled Network Slicing with Slice Isolation

One of the common characteristics of the previously mentioned studies is that they did not take the isolation of resources into account during slicing which is also one of the key expectations of network slicing. Isolation can be defined as the ability to ensure that congestion, attacks, and lifecycle-related events on one network slice does not negatively impact other existing slices. In this respect, the use of a shared infrastructure makes isolation a key requirement of network slicing. Thus, slice isolation can further enhance end user experience for use cases that require low latency and high bandwidth for optimized device and application performance.

Aiming to provide isolation among different services, Ref. [23] proposed a MEC slicing framework that allows network operators to instantiate heterogeneous slice services (e.g., video streaming, caching, and 5G network access) on edge devices. They pointed out that this is a challenging process as MEC resources are usually coupled. Hence, slicing one resource may lead to a performance degradation of another resource. To address this problem, they mathematically modelled coupling relationships among networking, storage and computation resources at each edge node by using collateral functions. Basically, collateral functions reflect whether the necessary resources are coupled and also determine the portion of different resources needed to be allocated on a given edge node. Their results are promising in a sense that they are able to instantiate slices six times more efficiently then state-of-the-art MEC slicing algorithms. Furthermore, their experimental results showed that they can simultaneously provide highly-efficient slicing of LTE connectivity, video streaming over WiFi, and video transcoding.

Ref. [34] posited that enabling resources isolation among slices is one of the most significant challenges for realizing softwarized base stations based on MEC. In this study, their aim is to isolate low latency slices as the primary concern and usage area of MEC is to enable low-latency applications. In this respect, they proposed an MEC-enabled slicing framework and proposed a novel slicing method for improving the isolation of low latency slices from the others. Their evaluation results showed that the proposed method can achieve minimal latency, even with a competing low-latency slice.

Ref. [35] took the isolation issue one step further and proposed a wireless network slicing solution considering both inter-slice isolation (i.e., no interference among slices), and intra-slice isolation (i.e., no interference between users in the same slice). In particular, they pointed out that in presence of selfish agents or data greedy applications, it is important for infrastructure providers to design an appropriate incentive plan to achieve and protect the social efficiency. Indeed, they use bidding algorithm where each slice, requested by network operators, needs to submit an individual bidding value. After receiving bids from different operators, the provider allocates resources to each bidding agent proportionally to their bidding values. After that, to provide fairness, they analysed optimal resource allocation under the Nash equilibrium and introduced a penalty value for each bidder. They mentioned that the key benefit of using such a resource bidding and allocation framework is that the network infrastructure providers do not know the true valuation of the network operators. Thus, isolation and fairness between slices can be provided by their proposed problem formulation. The proposed scheme achieves performance gain up to 13% in comparison to the equal sharing mechanism.

Specifically for the mMTC use case, Ref. [36] proposed an isolated network slicing approach for IoT applications. They considered the use case of a smart city required several different applications including air quality, temperature, traffic monitoring sensors, and smart buildings devices. As utilizing such IoT infrastructure can be costly, to reduce the market entry barriers and to efficiently exploit the potential of IoT devices, they proposed a scheme enabled by network slicing and virtualization technologies that enables the sharing of the IoT infrastructure between multiple isolated tenants. They concluded that the proposed NFV slicing orchestration mechanisms enabled the sharability of IoT resources and the automated life-cycle management of the IoT Slices.

There are also some studies focused on E2E slicing together with isolation of resources. As an example, Chien et al. proposed an E2E slicing framework for both computing and communication resources across the MEC architecture [37]. They pointed out that, when deploying E2E slicing in a MEC platform, the problem of isolating and allocating the computing resources in the central office and mobile edge, and network resources in the transport network and RAN must be carefully considered. As a result, they provided an NFV-enabled MEC infrastructure where a controller periodically reports the network resource status to the resource orchestrator agents which then maps this information to a physical resource to create a slice for the corresponding service. They showed that their framework successfully isolates the 5G resources between slices and ensures that resources of the deployed slices are merely sufficient to meet the latency requirements of the tenants. They also proved that the edge resources are very critical and crucial for the necessary computing and communication resources of URLLC and mMTC services.

Similarly, Ref. [38] designed an NFV-based MEC to realize E2E slicing over heterogeneous wireless networks. In particular, they focused on providing guaranteed E2E bandwidth and isolated resources to be able to serve applications that have QoS guarantees. Here, for big data related applications, QoS guarantees could be a high throughput or a delay-sensitive application, or it could be low latency. Specifically, the problem they focused on is that, when UEs connect to a wireless network and share a media, it would be difficult to ensure E2E QoS guarantees, especially for bandwidth hungry applications such as services that require video streaming. They pointed out that most of the current solutions do not rely on edge slicing, need modifications on devices, and have compatibility issues. In this respect, they provided the first solution that can create application-aware slices and provide QoS guarantees through the orchestration of MEC-enabled slices, without requiring any modifications of UEs.

Ref. [39] also focused on E2E slicing and proposed a novel E2E network slicing framework for IoT services in 5G systems. Mainly, their solution is the first real implementation of an IoT slicing framework integrated with a 5G core network. As part of this study, they provided the complete slice building procedure, as well as its integration with the OpenAirInterface and the OpenBaton orchestrators. They concluded that their slicing framework permits to host a high number of IoT devices with diverse QoS requirements compared to non-sliced networks.

Comparing the isolated and non-isolated methods, isolated slices offer guaranteed QoS and QoEs. In particular, as the slice isolation enables slices to be able to operate without any interference, preserving the isolation guarantees the performances as well as maintains the fairness between slices.

To summarize, an overview of all previously mentioned edge slicing studies is listed in the Table 2.

**Table 2.** Overview of edge-enabled slicing studies mentioned in Section 2.

| Ref. | Methodology | Objective | E2E | Slice Isolation | Outcome |
|------|-------------|-----------|-----|-----------------|---------|
| [27] | NFV | Scalable E2E service slicing | Yes | No | Lower latency for various web services |
| [29] | SDN & NFV | Resource utilization and load balancing | No | No | Up to 48% higher resource utilization than SOTA algorithms |
| [30] | SDN & NFV | Flexible and intelligent traffic steering | No | No | Up to 50% higher network throughput than SOTA algorithm |
| [31] | - | Maximize operator's average revenue | No | No | Flexible balance between the revenue and the average delay |
| [32] | NFV | Improve network slice acceptance ratio | Yes | No | Improved load balancing and reduced load of edge data centers |
| [23] | - | Efficient and scalable network slicing | No | Yes | Instantiates slices 6 times more efficiently than SOTA algorithms |
| [34] | NFV | Isolation of low latency slices | No | Yes | Maintains the same minimal latency with a competing low-latency slice |
| [35] | - | Efficient resource allocation | No | Yes | Nearly optimal bandwidth allocation |
| [36] | NFV | Automated E2E slicing | No | Yes | Lower E2E instantiation times for mMTC slices |
| [37] | NFV | Automated and flexible dynamic resource allocation | Yes | Yes | Guaranteed latency |
| [38] | SDN& NFV | Application-aware E2E slices on demand | Yes | Yes | Guaranteed E2E bandwidth |
| [39] | NFV | Flexible and dynamic placement of micro-services | Yes | Yes | Guaranteed QoS for migrating application flows |
| [33] | - | Achieve QoS goals of slices | Yes | - | Assigns the necessary number of C-RAN and MEC cores |

## 3. Integrating Machine Learning into Edge Sliced Networks

To analyse large amounts of data and obtain useful information for detection, classification, and prediction of future events, machine learning techniques are often applied in a lot of different applications and services. As it enables fast inference and autonomy within the networks, it is also envisioned to be an important component of 5G networks and beyond. An example of how ML algorithms can be integrated into edge-enabled network slicing architecture is presented in Figure 2.

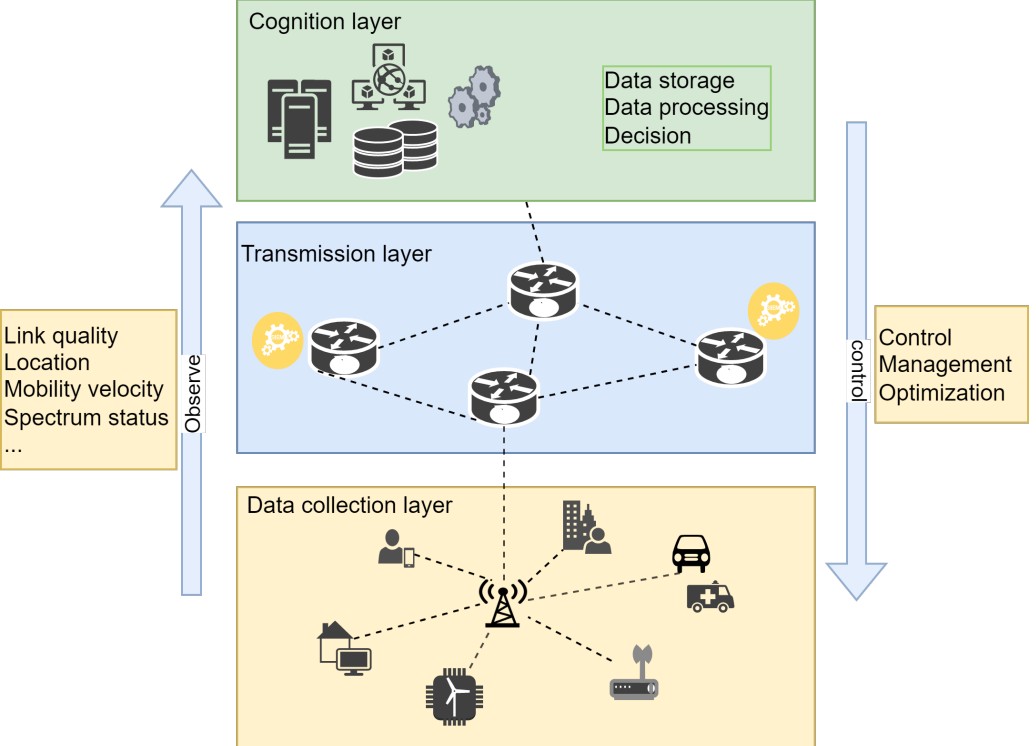

**Figure 2.** Integration of ML into wireless network architecture.

### 3.1. Machine Learning Applied to Network Slicing

As mentioned in Section 1, network slicing in 5G networks is realized by creating self-contained logical networks consisting of a combination of dedicated and shared resources depending on the needs and demands of each request. In order to efficiently perform construction and management of network slices, a large amount of data generated by mobile device users and vertical industries that are using the emerging services such as automotive communication, AR/VR, and remote healthcare needs to be examined. As it is complex for human beings to handle such high volumes of data in a limited time, one can utilize the machine learning for analysing the data as well as automating the network slicing processes. In particular, the knowledge and patterns we received from ML algorithms can provide us insight on how slices should be created to efficiently and effectively to utilize the edge resources.

There are several works in the literature taking advantage of ML during the slicing process. As an example, Ref. [40] developed a novel ML-based scheme for dynamic resource scheduling for networks slicing, aiming to achieve automatic and efficient resource optimization and E2E service reliability. As it is difficult to obtain user data due to privacy concerns, they used RL to extract knowledge by interacting with the network itself. In addition, aiming to improve the feature extraction of the RL framework and automate the decision-making process for resource allocation, the authors used Convolutional Neural Networks. Their experiment results showed that the proposed scheduler outperforms the heuristic, best-effort, and random approaches.

Ref. [41] also utilized ML in their study. Considering the fact that data-driven decisions may accelerate the performance of 5G networks, they proposed a Deep Learning Neural Network algorithm that analyses the overall traffic pattern, predicts future traffic, and is able to handle network load. In particular, the three main goals of their study were to effectively create slices, to improve slice creation and allocation processes by predicting traffic, and to be able to respond network failures by adapting the slices. To achieve that, they used random forest algorithm along with deep learning algorithm. Mainly, their algorithm is able to distribute incoming applications to the appropriate slices among eMBB, URLLC, mMTC, or master slices depending on the load and the outputs of their model. As a result, they achieved 95% slice prediction accuracy, i.e., whether the given device is a type of eMBB, URLLC, or mMTC, for unknown devices.

Similarly, Ref. [42] designed an RL-based network slicing framework. Mainly, they utilized three network slicing blocks which are a forecasting block that predicts the future traffic based on past information, a slicing admission control block, and a slicing scheduler block to meet the agreed SLAs and report back deviations to the forecasting block. With the help of RL, they are able to learn contemplated traffic models and heterogeneous mobility within slices. As part of this work, isolation between slices, allocation of resources, and the admission of resource requests by network slice tenants are also considered. Their experimental results revealed that the proposed forecasting technique boosts the system utilization up to 100% while reducing the computational time by 20% compared to non-forecasting algorithm.

Ref. [43] pointed out that there are still some conflicts on whether to use conventional ML algorithms or to use combination of ML and deep learning for efficient network slicing. To overcome this issue, they compared different techniques and report the performances. Mainly, they proposed an algorithm that combines ML and deep learning which involves three main phases: (1) data collection such as UE type, duration, packet loss ratio, bandwidth, delay rate, speed, jitter, etc.; (2) optimal weighted feature extraction; and (3) slicing classification to classify the network slices as eMBB, mMTC, or URLLC by a hybrid classifier using deep neural networks. As part of their experimental process, they compared their algorithm with other ML algorithms and proved that the combination of Ml and deep learning algorithms outperforms the conventional ML models by up to 45%.

Ref. [44] presented a conference paper on an ML aided network slicing algorithm which predicts if a service provider will be able to fulfil a new network slice request given

the conditions of the channel and the allocated resources. The main aim of this study is to predict the channel conditions in the near future using the Long Short-Term Memory algorithm. Experimental results showed that their DL based algorithm is able to reduce the number of false positive allocations by 75%.

Ref. [45] also focused on integrating ML algorithms, SDN, and NFV to build a comprehensive 5G architecture and an experimental framework. In this study, the authors proposed an approach for clustering and classifying mobile applications at an early stage, then each application is preserved a relevant bandwidth controlled by an SDN controller to guarantee that the network is working at high efficiency. The proposed framework is very promising as it is able to classify a high-quality YouTube video successfully to the correct slice, i.e., selected cluster. It also includes applications that require low latency and high bandwidth and, as a result, the UE played the video smoothly without any dropped frames.

Taking current gaps on 5G experimental prototypes into account, Ref. [46] proposed a 5G micro-service-based prototype that is able to auto-configure radio resources for network slices with ML. Mainly, the authors focused on eMBB and mMTC types of slices. To orchestrate the slicing mechanism and detect slicing ratio, they used ML-based forecasting. That is, once the training of ML model is completed, their algorithm dynamically provisioned the optimal slicing ratio and created new slices accordingly. As a result, they were able to increase the throughput by approximately 30% compared to the no forecasting case. However, they also pointed out that, this automation comes at the cost of increased utilization of CPU of the host system.

Ref. [47] dealt with the prediction and management issues of vehicular use cases in the existing research. Having these issues in mind, they proposed an ML-based resource allocation strategy for vehicular network slicing. Their algorithm works as follows. Firstly, a Convolutional Long Short-Term Memory algorithm is used to analyse temporal and spatial dependencies of service traffic. Then, to keep resource management in accordance with user mobility, a new wireless resource management scheme is used. Finally, the resource allocation algorithm based on the primal dual interior-point method is used to solve the optimal slice weight allocation. Their results showed that the proposed method can successfully predicts the future service traffic and is able to adjust the slice weights accordingly.

Ref. [48] also studied vehicle use case in 5G networks. In particular, they focused on electrical vehicles by trying to assign them to charging stations with minimum collision and maximum usage of the network. In order to maintain all the requests, in addition to network slicing, they also utilized an unsupervised ML algorithm. To measure the performance of their algorithm, they conducted several experiments and concluded the that proposed algorithm can allocate resources more efficiently and hence can serve more vehicles compared to a first-come-first-served approach.

Similarly, Ref. [49] proposed an intelligent network slicing architecture for vehicular communication services using NFV and ML algorithms. Mainly, their algorithm composed of four layers which are a network infrastructure virtualization layer, which virtualizes resources from a dedicated hardware; an intelligent control layer, which determines patterns of vehicular networks and performs self-configuration of network slices; a network slice layer, which defines different slices based on QoS requirements; and a service layer which captures the QoS requirements. In this work, they used Deep RL in combination with convolutional networks as well as a Long Short-Term Memory algorithm. This is because their data is stochastic but also has many dimensions. They concluded that the proposed framework is promising as it can improve QoS by allowing the control layer to make decisions according to historical observations.

Focusing on the Industrial IoT application, Ref. [50] proposed a dynamic slicing method to guarantee QoS by isolating urgent traffic and eliminating resource starvation. To do that, they utilized an online learning method that clusters devices to the most appropriate slices using Online Gaussian Mixture Model. Their simulation results highlighted the efficiency of the proposed method in avoiding resources starvation and providing QoS for devices while respecting the latency requirements and decreasing the energy consumption.

Different to the above studies, Ref. [51] presented a fairly new approach. In particular, to optimize load balancing further, they designed an ML-based network sub-slicing model, where each logical slice is also divided into slices, called as sub-slices. Considering the fact that each network slice needs to fulfil several different requirements, such as low latency, high reliability, and high spectral efficiency, their aim was to create sub-slices where each sub-slice focuses on a single requirement. As an example, one sub-slice focuses on spectral efficiency, whereas the other focuses on providing low latency with reduced power consumption. In particular, to identify requirements of different applications, they utilized a Support Vector Machine algorithm. In addition, to create clusters of sub-slices for grouping similar types of application services, they used K-means algorithm. Experimental results showed that the proposed algorithm outperforms the state-of-the-art algorithms in terms of improved performance and reduced energy consumption.

Considering the above applications, it can be seen that ML is a powerful and commonly used tool for network slicing as it yields improved performance and faster convergence in network management automation. That is, due to the complexity of the performed tasks which address different purposes, network slice controllers, being responsible for the monitoring and management of the functionalities of the overall system, can use the feedback and results obtained from ML algorithms. In particular, for network resource management in large-scale systems, ML-based slicing methods can be used by network controllers to solve the joint allocation problem of communication, caching, and computing resources due to its ability to handle complicated, dynamic, and heterogeneous features.

### 3.2. Machine Learning Applied to Edge Computing

As mentioned in Section 1, machine learning models are built from the collected data to enable the detection, classification, and prediction of future events. Due to bandwidth, storage, and privacy concerns, it is often impractical to send all the data to a centralized location. As the edge servers have limited storage resources and processing capabilities, using machine learning algorithms can optimize the performance of these servers. In this section, we will present a comprehensive literature review on ML applications in edge computing systems.

Utilizing a deep neural network model, Ref. [52] proposed an offloading mechanism for UEs to offload computational intensive tasks to the edge servers, as running such tasks on UEs can limit the potential of Ml models and reduce their accuracy. In this respect, they proposed an ML based scheduling and placement algorithm. More specifically, their proposed solution automated the orchestration and deployment of edge applications while guaranteeing efficient usage of resources, scalability, and fault-tolerance of the network. Their experimental results showed a significant gain compared to cloud-based offloading strategies in terms of accuracy and latency.

Sun et al. focused on an Industrial IoT case. Different than the existing works that are aimed to improve either power efficiency or the latency of industrial IoT devices, in this work, the authors also focused on improving the service accuracy. Mainly, they used DL-based transfer learning for image recognition tasks and for offloading decisions [53]. As a result, they showed that AI-enabled edge servers could serve more traffic compared to AI-enabled cloud servers.

Ref. [54] worked on a vehicular use case. In their work, they focused on how to select which node on which to offload the computationally intensive tasks to make efficient offloading decisions and proposed an ML-based vehicular edge orchestrator. Their algorithm is based on a classifier which detects whether the offloading decisions are successful. In addition, a regression model estimated the service time of all the offloading decisions. Based on these two outcomes, an edge node that has the lowest service time is selected. In this respect, to identify the most successful ML algorithm, they experimented with several ML algorithms such as naive Bayes, support vector machine, etc. One of the key contributions of their work is that they evaluated their algorithm in a realistic setting and proved that their algorithm outperforms random simple moving average based, multi-armed bandit

theory based, and game theory based vehicular edge orchestrators in terms of the task failure rate and service times.

In their paper, Ref. [55] evaluated the performance of three machine learning algorithms which are k-nearest neighbour, support vector machine, and naive Bayes, running on the edge servers. In particular, they utilized online ML tasks where patterns are extracted from continuous streams of data and the learned models evolved over time to capture complex relationships among many different variables. They stated that, for energy consumption, support vector machine has the largest reward, whereas naive Bayes has the lowest. In fact, their evaluation results are very promising as they are also able to prove the benefits of using ML in the edge nodes on energy consumption.

Aiming to provide a scheduling mechanism in real time to realize intelligent cognitive assistant applications, Ref. [56] proposed a novel RL based task assignment approach for the edge servers. As these applications work in real time, especially for healthcare scenarios, they formulated an RL program to reduce the running time of the assignment task itself. Their simulation results showed that they are able to reduce the task processing time by 13–22% compared to other existing methods.

Ref. [57] investigated a joint task, spectrum, and power allocation problem for wireless networks equipped with MEC servers. Mainly, they formulated an optimization problem to minimize the maximal computational and transmission delay among all users. To solve this problem, they utilized an RL algorithm. Compared to the conventional RL algorithms, their algorithm is able to record historical resource allocation schemes and users' states to avoid learning the same information, thus improving the convergence speed and the learning efficiency. Thus, they are able to reduce the delay among all users up to 18% compared to the standard Q-learning algorithms.

There are also some studies that handle the ML-based edge computing systems from the point of view of security. As an example, trying to secure communications and actively detecting unknown attacks, Ref. [58] proposed a combined deep and unsupervised learning model for the MEC environment. Basically, their algorithm consists of a pretraining phase that relies on unsupervised learning to detect hidden units and a fine tuning phase using deep learning to fine tune the parameters that are already trained in the pretraining phase. They compared their model with four other ML-based algorithms, where their algorithm stood out from others based on its ability to capture the nonlinear relations between attacks and corresponding features, and its multi-stacked modules that calculate the nonlinear mapping between input and output. The results showed that their model is able to improve the overall accuracy by 6%.

Trying to reduce data traffic and latency for IoT devices, Ref. [59] explored merging ML and edge and cloud computing. They stated that the reason they used deep learning is that it is able to transform data into hierarchical abstract representations that are beneficial for IoT data analytics. They deployed the encoder part of the trained model on the edge to reduce dimensionality, whereas the decoder was employed on the cloud to reconstruct the original signal. As a result, they showed that their method outperforms other deep learning solutions for IoT devices. In addition to that, they grouped the sensors according to location and made comparisons with non-grouped cases, showing their method can reduce network traffic up to about 80%.

In short, ML assisted edge computing reduces latency as well as avoids high computing costs by processing the data at the edges rather than in centralized clouds. In addition, it also empowers new technologies such as autonomous vehicles and medical devices by enabling real-time feedback and results that are critical for these services.

### 3.3. Machine Learning Applied to Edge-Enable Network Slicing

As the emerging 5G services have highly diverse performance requirements such as high bandwidth, low latency, high reliability, etc., efficient accommodation of those needs requires the use of edge-enabled slicing together with ML techniques. In this respect, there are a couple of studies in the literature that employ ML algorithms in edge-enabled

slicing frameworks. As an example, Ref. [60] proposed a resource orchestration method for wireless edge networks based on decentralized deep RL and network slicing techniques. In particular, the deep RL agent learns the resource demands of network slices and orchestrates the resource allocation accordingly. The authors stated that, due to the temporal and spatial dynamics of the slice traffic and the complex tradeoffs between the performance of network slices and the resource orchestration, it is inefficient to use a centralized learning agent to orchestrate resource allocations to network slices. Additionally, a centralized learning agent needs to obtain network performance data from all the network nodes, which introduces excessive communication overhead and delay. Thus, they adopted a decentralized deep RL approach to automate dynamic E2E network slicing and to optimize the performance of the slices under the constrained networking and computing resources. Simulation results showed that their orchestration agent is able to autonomously orchestrate E2E resources under varying slice traffic.

Considering data-intensive and latency-sensitive tasks, Ref. [61] proposed to extend network slices to aerial vehicles equipped with MEC nodes. This is because when data are produced by ground devices they may not able to reach the edge servers in time. Particularly, this model consisted of a RL agent that analyses the environment and decides whether to offload jobs to aerial vehicles based on power consumption, job loss, and incurred delay. Experimental results showed that this proposed method can work at runtime with a great flexibility.

Combined with edge computing and network slicing, ML based solutions enable autonomous slice management, control, orchestration, and optimization. In this respect, one of the main advantages of ML is to accurately classify types of applications for automating the network slicing process. In addition to that, ML models can also be useful for optimizing resource allocation in edge enabled network slicing, particularly for predicting, assigning, and optimizing the limited edge resources.

An overview of previously mentioned ML assisted studies is presented in Table 3.

**Table 3.** Overview of studies that combine ML, network slicing, and edge computing techniques mentioned in Section 3.

| Ref. | Application Area | Objective | Methodology | Outcome (Compared to SOTA Methods) |
|---|---|---|---|---|
| [47] | Network Slicing | Resource allocation for vehicles | Convolutional Long Short-Term Memory | Reduced transmission and waiting delay by 15.33 ms |
| [45] | Network Slicing | Comparison of ML algorithms for network slicing | Naive Bayes /SVM /Neural Network /RF | Classification accuracy of different ML methods |
| [51] | Network Slicing | Optimize network load balancing problems | SVM /K-means | Improved latency, load balancing, and power efficiency |
| [50] | Network Slicing | Dynamic slicing and resource allocation for Industrial IoT | Online Gaussian Mixture Model | Decreased energy and packet error rate |
| [46] | Network Slicing | Auto-configure radio resources | Regression Model | Increased throughput |
| [42] | Network Slicing | Enhance performance of network slicing | Regression Model | Increased system capacity |
| [48] | Network Slicing | Efficient communication btwn the vehicle and charging station | Unsupervised Learning | Increase the throughput and decreased latency |
| [41] | Network Slicing | Optimize network load and resources | Deep Neural Network | Predicts the most accurate network slice for an application |
| [49] | Network Slicing | Comparison of network slicing methods for V2X services | Deep Reinforcement Learning | Improved packet latency and Block Error Rate |
| [40] | Network Slicing | Automated resource optimization and E2E service reliability | Deep Reinforcement Learning | Dynamically allocated resources within QoS requirements |
| [44] | Network Slicing | Predict a new network slice request can be fulfilled or not | Deep Learning | Reduced the number of false positive allocations by a 75%. |
| [43] | Network Slicing | Classify slices as "eMBB, mMTC, or URLLC" | DNN and Unsupervised learning | Increase accuracy up to 1.19% |
| [55] | Edge Computing | Smart healthcare applications | K-nearest neighbours /Naive Bayes /SVM | Reduced energy consumption and computation time |
| [54] | Edge Computing | Efficiently operate self-driving vehicles | Naive Bayes /SVM /Multilayer Perceptron | Decreased task failure rate and service time |
| [59] | Edge Computing | Reducing network traffic and latency | ML | Reduced network traffic up to 80% |
| [53] | Edge Computing | Increase service accuracy | Transfer Learning | Serve more traffic up to 10% |
| [58] | Edge Computing | Improve communications security | Deep Learning | Improved classification accuracy by 6% |
| [52] | Edge Computing | Offloading mechanism for edge devices | Deep Neural Network | Improved accuracy and latency |
| [56] | Edge Computing | Develop Intelligent Cognitive Assistants | Reinforcement Learning | Reduced task processing time (22%) and energy consumption (10%) |
| [57] | Edge Computing | Minimize the computational and transmission delay | Reinforcement Learning | Reduced maximal delay up to 11.1% |
| [61] | Edge-enabled slicing | Minimize power consumption and delay for aerial vehicles | Reinforcement Learning | Improved performance, scalability, compatibility |
| [60] | Edge-enabled slicing | E2E resource orchestration | Deep Reinforcement Learning | Improved performance, scalability, compatibility |

It should be also noted that, in most of the systems with network slicing mechanisms, all the useful information gathered and analysed by ML algorithms are conveyed to network slice controllers which then manage slice life-cycles and configure necessary resources and functions. Due to the complexity of these tasks, a network slice controller can be composed of multiple orchestrators that are in charge of a subset of functionalities. Thus, to be able to ensure and maximize the benefits received from these solutions, it is important to ensure that the controllers are in coordination with each other about the status of the operations and maintaining a centralized view.

## 4. Proposed Solution

As mentioned earlier, while applying network slicing to heterogeneous networks with multiple domains, hierarchical SDN controller architecture is largely preferred. In such cases, different SDN controllers, each spanning a different domain, are needed to synchronize with the root controller to preserve the consistency and reliability within the network

as well as to further utilize the benefits of edge enabled network slicing mechanism. In this respect, despite various proposals on how to realize edge enabled network slicing, most of them overlooked the question of how exactly controllers should synchronize with each other and assumed that the logically centralized network view somehow existed.

Motivated by this idea and the benefits of ML, we propose a synchronization method for hierarchically distributed SDN controllers using RL aiming to improve the decisions of controllers on slice life-cycles, resource management, routing paths, and offloading mechanisms, towards more effective edge-enabled network slicing. In this respect, there are two main reasons why we believe that RL is a good candidate for the controller synchronization problem. The first reason is that, in wireless networks where several applications with human involvement exist, it is almost impossible to find labelled data for training the ML algorithms, whereas unlabelled data are often abundant and easily available [62]. Secondly, optimizing the synchronization process for heterogeneous networks is indeed a long-term decision that is affected by multiple conditions. Such problems are considered decision-making tasks where RL offers fast near-optimal solutions; this is also validated by other research in the literature [63,64].

Although there are a few existing works related to our research question, our solution differs from them in the following ways [65,66]: (i) As opposed to existing works where only some of the controllers are being synchronized at each time, our solution aims to synchronize all controllers at once. By doing that, we are expecting to have lower run time complexities as well as lower energy consumption in the long run compared to the partial synchronization method. (ii) Previous works assumed that the synchronization times are given. Contrarily, in our study, the more realistic case of detecting the synchronization times as well as the frequency is going to be explored.

An example system architecture is presented in Figure 3. In particular, the data that we are going to use in our RL solution are the number of users and devices, mobility of users, link quality, CPU load, and memory usage of each edge server. During implementation, we are also aiming to address several issues such as dependencies among different slices and resources and the security of individual controllers, as well as edge cases such as the procedures that we are going to apply in case of a main controller failure.

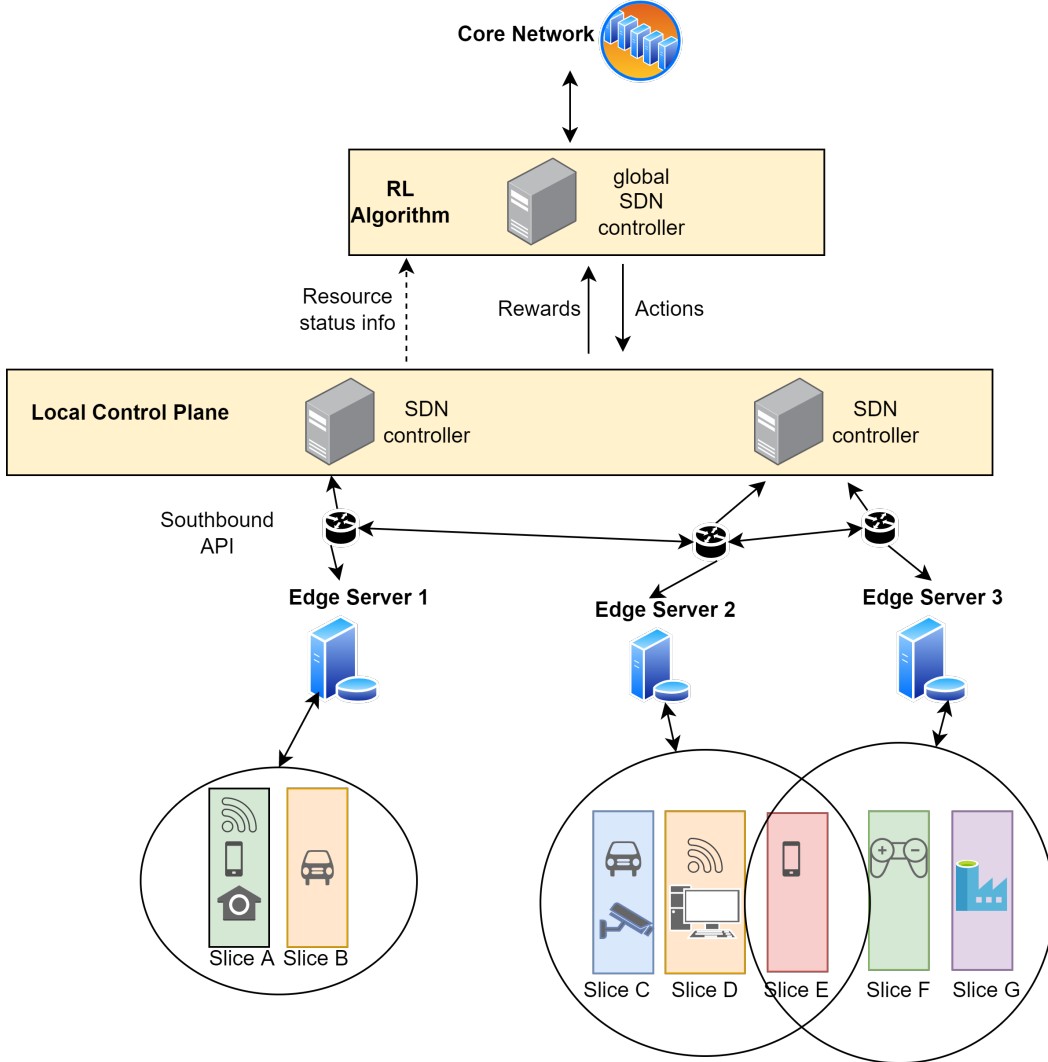

**Figure 3.** An example system architecture representing proposed research idea.

## 5. Conclusions

Network slicing and edge computing are key enabling technologies of 5G and beyond networks considering their ability to create scalable and flexible networks while meeting the QoS and SLA requirements of different applications. Recent studies also showed that slicing the edge resources reveals the network capabilities further by satisfying the stringent QoS and QoE requirements of emerging applications. In this respect, this review paper presented existing edge-enabled network slicing solutions, frameworks, and use cases as well as potential benefits. Moreover, we also provided existing studies on how to combine ML methods with these concepts. By doing that, we aim to guide the research with a comprehensive analysis on edge enabled network slicing mechanisms combined with ML methods. In addition, considering the fact that synchronization between the slicing controllers improves the slicing decisions, we also proposed an RL-based controller synchronization solution. In particular, we believe that our method will be useful in terms of optimal routing path detection, resource balancing, and data offloading decisions between both the edge servers and their slices towards more effective edge-enabled network slicing.

**Funding:** This research was partially funded by ECSEL-JU BRAINE grant number 876967 and NGF Quantum Delta CAT2.

**Institutional Review Board Statement:** Not applicable.

**Informed Consent Statement:** Not applicable.

**Data Availability Statement:** There are no relevant datasets presented in this article.

**Acknowledgments:** Authors would like to thank all BRAINE and CAT2 partners involved in this work.

**Conflicts of Interest:** The authors declare that the research was conducted in the absence of any commercial or financial relationships that could be construed as a potential conflict of interest.

## Abbreviations

The following abbreviations are used in this manuscript:

| | |
|---|---|
| MDPI | Multidisciplinary Digital Publishing Institute |
| ML | Machine Learning |
| DL | Deep Learning |
| RL | Reinforcement Learning |
| SDN | Software-defined networking |
| NFV | Network functions virtualization |
| MEC | Multi-access edge computing |
| RAN | Radio access network |
| QoS | Quality of service |
| QoE | Quality of experience |
| URLLC | Ultra-reliable low-latency communication |
| eMBB | Enhanced Mobile Broadband |
| mMTC | Massive machine type communication |
| IoT | Internet of Things |
| UE | User equipment |
| SVM | Support Vector Machine |
| RF | Random Forest |

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
