# Peer review of "Integration of Network Slicing and Machine Learning into Edge Networks for Low-Latency Services in 5G and beyond Systems"

_applsci, doi:10.3390/app12136617_

Round 1

Reviewer 1 Report

In this paper, the authors provided a comprehensive study on edge-enabled network slicing frameworks and potential solutions with example use-cases. Some suggestions are presented as follow.

1. As a very timely topic in the communication networks, there are so many paper presenting network slicing. The authors are suggested to highlight the main difference between this paper and other published works.

2.  Machine learning used in network slicing and edge computing is very attractive. However, how to guarantee the precision, such channel estimation, delay-sensitive applications, are very challenging and cannot be ignored in practice. The authors are suggested to discuss or to offer a future direction.

Author Response

Dear reviewer, Thank you for your feedbacks. Please find my corrections based on your comments and suggestions:

  1. The main difference between my work and the previous works on network slicing and edge computing is that, my work focuses on their combination and reviews the most recent studies on edge enabled network slicing mechanisms by also including current machine learning applications in this field. Please fin the highlighted difference as a short paragraph at the introduction section ( between the lines 180-186).
  2. Thank you for your suggestion. As I would like to not include such applications of machine learning to not make the paper more complicated, I included these points as a short paragraph and discussed the current applications. Please find them under the Introduction/Machine Learning section (between the lines 158-164).

Reviewer 2 Report

In this paper, the authors propose a network slicing and machine learning integration method into edge networks to improve latency for 5G and beyond systems. In the reviews' opinion, this research issue is very interesting and attractive. However, the authors should improve to enhance the quality of the paper, as follows.

1) In line 224, present "[23] introduced a 5G", in my opinion, the authors should re-write "In [23] introduced a 5G" Similarity, in lines 240, 257, 275, 315, 322.

2) In Section 1, Introduction, the authors should mention the key challenges that 5G are facing such as performance, energy-saving and security and privacy. It highlights the timely and attraction of this work. These issues are presented in the recent research, which the author should reference: Wireless Communication Technologies for IoT in 5G: Vision, Applications, and Challenges, Wireless Communications and Mobile Computing.

3) In Table 3, the authors present an overview of studies that combines ML, network slicing, and edge computing techniques. In my opinion, the authors should more detail explain and analyze to make it convenient for readers

4) The authors should move Figure 3 (the proposed model) into Section 4.

Author Response

Dear reviewer, Thank you for your feedbacks. I have updated my manuscript based on your comments and suggestions as below:

  1. Thank you for your correction.
  2. Thank you for pointing out the current critical challenges faced by 5G and beyond networks. For both network slicing and machine learning sections, I now included such issues as a short paragraph and discussed the important points. Please find them between the lines (74-81) and (119-124).
  3. Please find the new version of Table 3. I try to improve the readability of the table as much as I am able to.
  4. Thank you for this correction.

Round 2

Reviewer 2 Report

I am agreed with the reception and explanations of the authors in the revised version.

Author Response

Thank you for your time and comments